# The Effect of Prohibitins on Mitochondrial Function during *Octopus tankahkeei* Spermiogenesis

**DOI:** 10.3390/ijms241210030

**Published:** 2023-06-12

**Authors:** Jingqian Wang, Xinming Gao, Chen Du, Daojun Tang, Congcong Hou, Junquan Zhu

**Affiliations:** 1Key Laboratory of Aquacultural Biotechnology, Ningbo University, Ministry of Education, Ningbo 315211, China; wangjingqian0815@163.com (J.W.); nbugxm4851@163.com (X.G.); 8788182@163.com (C.D.); tangdaojun@nbu.edu.cn (D.T.); houcongcong@nbu.edu.cn (C.H.); 2Key Laboratory of Marine Biotechnology of Zhejiang Province, Ningbo University, Ningbo 315211, China

**Keywords:** prohibitins, mitochondrion, *Octopus tankahkeei*, spermiogenesis

## Abstract

Mitochondria are essential for spermiogenesis. Prohibitins (PHBs; prohibitin 1, PHB1 or PHB, and prohibitin 2, PHB2) are evolutionarily conserved and ubiquitously expressed mitochondrial proteins that act as scaffolds in the inner mitochondrial membrane. In this study, we analyzed the molecular structure and dynamic expression characteristics of *Ot*-PHBs, observed the colocalization of *Ot*-PHB1 with mitochondria and polyubiquitin, and studied the effect of *phb1* knockdown on mitochondrial DNA (mtDNA) content, reactive oxygen species (ROS) levels, and apoptosis-related gene expression in spermatids. Our aim was to explore the effect of *Ot*-PHBs on mitochondrial function during the spermiogenesis of *Octopus tankahkeei* (*O. tankahkeei*), an economically important species in China. The predicted *Ot*-PHB1/PHB2 proteins contained an N-terminal transmembrane, a stomatin/prohibitin/flotillin/HflK/C (SPFH) domain (also known as the prohibitin domain), and a C-terminal coiled-coil domain. *Ot-phb1/phb2* mRNA were widely expressed in the different tissues, with elevated expression in the testis. Further, *Ot-*PHB1 and *Ot-*PHB2 were highly colocalized, suggesting that they may function primarily as an *Ot*-PHB compiex in *O. tankahkeei*. *Ot-*PHB1 proteins were mainly expressed and localized in mitochondria during spermiogenesis, implying that their function may be localized to the mitochondria. In addition, *Ot-*PHB1 was colocalized with polyubiquitin during spermiogenesis, suggesting that it may be a polyubiquitin substrate that regulates mitochondrial ubiquitination during spermiogenesis to ensure mitochondrial quality. To further investigate the effect of *Ot*-PHBs on mitochondrial function, we knocked down *Ot*-*phb1* and observed a decrease in mtDNA content, along with increases in ROS levels and the expressions of mitochondria-induced apoptosis-related genes *bax*, *bcl2*, and *caspase-3* mRNA. These findings indicate that PHBs might influence mitochondrial function by maintaining mtDNA content and stabilizing ROS levels; in addition, PHBs might affect spermatocyte survival by regulating mitochondria-induced apoptosis during spermiogenesis in *O. tankahkeei*.

## 1. Introduction

Prohibitins (PHBs) are high-molecular-weight circular heterodimeric complexes comprising 12 to 16 PHB1-PHB2 units that predominantly localize in the inner mitochondrial membrane (IMM) and influence its function and kinetics. PHB1 and PHB2 share over 50% similarity and are evolutionarily conserved across all phyla [1]. Studies on the domains of PHBs have revealed that both PHB1 and PHB2 contain the stomatin/prohibitin/flotillin/HflK/C (SPFH), C-terminal coiled-coil, and N-terminal transmembrane domains [2,3]. The SPFH domain is an evolutionarily conserved PHB domain that is common to several other scaffold proteins (including stomatin, flotillin, and HflK/C). The C-terminal coiled-coil domain represents the interaction domain between PHB1 and PHB2, and the N-terminal hydrophobic domain localizes PHBs to the IMM [4]. PHBs have multiple functions in the IMM; for example, they interact with mitochondrial M-AAA proteases to degrade mitochondrial membrane proteins [5], bind with mitochondrial DNA (mtDNA) to stabilize the mitochondrial genome [6], stabilize mitochondrial complex I protein components to influence mitochondrial reactive oxygen species (mtROS) generation [7], and affect mitochondrial fusion, playing an essential role in maintaining mitochondrial morphology [8]. Furthermore, PHBs also regulate mitochondrial ubiquitination as substrates [9].

As the “energy factories” of cells, mitochondria are intricately linked to spermiogenesis, which is a highly complex but systematic transformation of spermatids into mature sperms [10,11,12]. The morphology, gene expression, protein composition, and migration of mitochondria during spermiogenesis play a role in the supply of metabolic energy to spermatids at various developmental stages, such as acrosome formation and nuclear shape changes, as well as in the storage of energy for sperm motility [13]. Research indicates that mitochondrial structural abnormalities, such as abnormal expression of mitochondrial membrane proteins, mtDNA deletion or mutation, abnormal electron transport chain components, excessive reactive oxygen species (mtROS) levels, and abnormal mitochondrial fusion, might result in poor sperm quality and motility [11,14,15].

The multiple functions of PHBs within IMM, the importance of mitochondria in spermiogenesis, and the role of PHBs in mitochondrial function during spermiogenesis have attracted extensive attention. For example, in *Eriocheir sinensis* [16], *Boleophthalmus pectinirostris* [17] and *Plestiodon chinensis* [18], PHB is mainly colocalized with mitochondria during spermiogenesis, indicating that PHBs are essential for mitochondrial function during spermiogenesis [14,19]. However, the specific function of PHBs in mitochondria during spermatogenesis remains unclear.

*Octopus tankahkeei* (*O. tankahkeei*) is a small marine organism belonging to the phylum Mollusca and class Cephalopoda. This species is an economically important aquatic species in China and its current market supply mainly depends on its capture in natural marine environments, since artificial seedling rearing and culture technologies are still in the exploration stage, and existing research on fertilization and developmental biology in this species is limited. Therefore, studying the sperm of *O. tankahkeei* can provide a theoretical basis for artificial breeding. Zhu et al. [20] studied mitochondrial dynamics during spermiogenesis. Their study revealed that in stage I of spermiogenesis, spermatids had well-developed mitochondrial cristae in the cytoplasm and abundant internal matrices (Figure 1a). In stage II, more mitochondria were present in the cytoplasm, generally concentrated at one end, and visible mitochondrial fusion was observed (Figure 1b). Mitochondria gradually migrated to the posterior nuclear end in stage III of spermiogenesis, (Figure 1c), and adhered to the nuclear membrane in stage IV (Figure 1d). In stage V, the migrated mitochondria formed the mitochondrial sheath in the midpiece of the sperm tail (Figure 1e). The formation of the mitochondrial sheath is an adaptation of sperm to internal fertilization. Compared with in vitro fertilization, mollusk sperm can store more energy to meet the needs of prolonged movement and fertilization in the female reproductive tract. Based on the high evolutionary conservation of PHBs and their importance in mitochondria and spermiogenesis, we hypothesized that PHB1 and PHB2 function as complex PHBs in *O. tankahkeei* and participate in spermiogenesis by affecting mitochondrial function.

## 2. Results

### 2.1. Biological Function Analysis of the Ot-phb1 Gene

The full-length *Ot-phb*1 cDNA sequence was 1094 bp long (GenBank number: HM627864.1) and had an 819 bp open reading frame encoding 273 amino acids (aa) (Figure 2a). The predicted molecular weight and isoelectric point of *Ot*-PHB1 were approximately 30.2 kDa and 5.69, respectively. *Ot*-PHB1 contains a C-terminal coiled-coil domain, an SPFH domain, and an N-terminal transmembrane domain with a coiled-coil in the SPFH domain (Figure 2c), and its possible protein tertiary structures were predicted using online tools (Figure 2d,f). Multiple sequence alignment findings revealed that *Ot*-PHB1 shared approximately 66.3–95.2% sequence identity with its homologs (Appendix A; Figure 2b). Furthermore, we found four conserved PHB1 lysine sites in all ten species, all of which were located in the SPFH domain, and two of which were simultaneously located in the coiled-coil domain. A phylogenetic tree was constructed to analyze the evolutionary relationship between *Ot*-PHB1 and PHB1 in other species (Figure 2f). *Ot*-PHB1 was found to have a close evolutionary relationship with PHB1 in *Octopus bimaculoides,* which also belongs to Cephalopoda. These results indicated that the *Ot*-PHB1 protein is highly conserved.

### 2.2. Biological Function Analysis of the Ot-phb2 Gene

The full length of the *Ot-phb*2 cDNA sequence was 1162 bp (GenBank number: KY807538) and had an 888 bp open reading frame encoding 296 aa (Figure 3a). The predicted molecular weight and isoelectric point of *Ot*-PHB2 were approximately 33.3 kDa and 9.96, respectively. Similar to those of *Ot*-PHB1, *Ot*-PHB2 proteins contain a C-terminal coiled-coil domain, an SPFH domain, and an N-terminal transmembrane domain, with the coiled-coil domain within the SPFH domain (Figure 3c). The prediction of *Ot*-PHB2 protein tertiary structures using online tools showed that the SPFH domains of *Ot*-PHB1 and *Ot*-PHB2 were similar (Figure 3d,f). Multiple sequence alignment revealed that *Ot*-PHB2 shared approximately 63.9–97.0% sequence identity with its homologs (Appendix A; Figure 3b). In the ten species including *O. tankahkeei*, we found six conserved lysine sites of PHB2 all located in the SPFH domain, with three of them simultaneously located in the coiled-coil domain. A phylogenetic tree was used to analyze the evolutionary relationship between *Ot*-PHB2 and PHB2 in other species (Figure 3f). *Ot*-PHB2 had a close evolutionary relationship with PHB2 in the cephalopod, *Octopus bimaculoides*, although the evolutionary relationship of PHB1 and PHB2 with vertebrates, such as mammals, birds, reptiles, and teleost fishes, was found to be distant. These results indicated that the *Ot*-PHB2 protein is also highly conserved.

### 2.3. Tissue Expression Analysis of PHBs

qPCR was used to detect the expression of the *Ot*-*phb1* (Figure 4a) and *Ot-phb2* (Figure 4b) in the different tissues of *O. tankahkeei*. The results showed that both *Ot*-*phb1* and *Ot*-*phb2* were expressed in the muscles, testes, rectum, hepatopancreas, gill, salivary glands, stomach, kidney, caecum, craw, and heart, with the highest expression in the testes.

### 2.4. Ot-PHB1 and Ot-PHB2 Function as a PHB Complex during O. tankahkeei Spermiogenesis

Sequence alignment of *Ot*-PHB1 and *Ot*-PHB2 proteins using a dot matrix revealed that they were highly homologous subunits with approximately 54.15% amino acid sequence identity (Figure 5a). IF results showed that *Ot*-PHB1 and *Ot*-PHB2 protein signals were detected and colocalized during spermiogenesis (Figure 5b). In the early stages of spermiogenesis, *Ot*-PHB1 and *Ot*-PHB2 were highly colocalized in the perinuclear cytoplasm, and during spermatid development, they were also colocalized in the midpiece of sperm. These results suggest that PHB1 and PHB2 may bind to each other to form a heterodimeric PHB complex involved in the spermiogenesis of *O. tankahkeei* (Figure 5c).

### 2.5. The Relationship between Ot-PHBs and Mitochondria in the Spermiogenesis of O. tankahkeei

WB revealed that *Ot*-PHB1 and *Ot*-PHB2 were mainly expressed in the mitochondria of spermatids (Figure 6), but *Ot*-PHB2 was also found to be expressed in cells excluding mitochondria, which may be related to other functions of the PHB2 monomer. We subsequently confirmed the relationship between *Ot*-PHBs and mitochondria in spermiogenesis using IF (Figure 7). The findings revealed that *Ot*-PHB1 and mitochondrial signals co-localized during spermiogenesis, first in the perinuclear cytoplasm and subsequently migrating to the midpiece of the sperm during spermatid development. This suggests that *Ot*-PHB1 may influence the physiological function of the mitochondria during spermiogenesis, indicating that *Ot*-PHBs may participate in *O. tankahkeei* spermiogenesis by influencing the physiological function of mitochondrial.

### 2.6. The Relationship between Ot-PHBs and Polyubiquitin in the Spermiogenesis of O. tankahkeei

As shown in Figure 8, *Ot*-PHB1 protein and polyubiquitin signals were detected during spermiogenesis in *O. tankahkeei,* were highly colocalized during spermiogenesis, and migrated to the midpiece of sperm during spermatid development. This suggests that *Ot*-PHBs may be substrates for polyubiquitin during the mitochondrial ubiquitination process in *O. tankahkeei* spermiogenesis.

### 2.7. Detection of phb1 RNAi Efficiency

qPCR and WB were used to detect the effect of RNAi on the expression of *phb1* mRNA (Figure 9a) and PHB1 protein (Figure 9b,c). The results showed that RNAi significantly reduced the expression of *phb1* mRNA and PHB1 protein, indicating that successful *phb1* knockdown can be used for follow-up studies.

### 2.8. Effect of phb1 Knockdown on ROS Levels

The H_2_O_2_ levels in the testis of *O. tankahkeei* were significantly increased after phb1 *knockdown* (Figure 10), indicating that phb1 affects mitochondrial ROS levels in in the testis of *O. tankahkeei*.

### 2.9. Effect of phb1 Knockdown on mtDNA Content

We used qPCR to amplify mtDNA using *cyt b* (a mitochondrial gene). The result showed that knockdown of *phb1* resulted in significantly decreased copy numbers of mtDNA in spermatids (*p* < 0.05) (Figure 11), indicating that PHB1 is critical for maintaining the normal mtDNA content during spermiogenesis in *O. tankahkeei*.

### 2.10. Effect of phb1 Knockdown on Apoptosis-Related Genes

Knockdown of *phb1* significantly increased the mRNA expression of proapoptotic gene *bax* (Figure 12a), and *caspase-3* (Figure 12c); the mRNA expression of the anti-apoptotic gene *bcl2* was more stable (Figure 12b).

## 3. Materials and Methods

### 3.1. Animals and Tissues

Mature male *Octopus tankahkeei* were purchased from the Ningbo aquatic market, and samples (approximately 200 mg) of the muscles, testes, rectum, hepatopancreas, gills, salivary gland, stomach, kidney, blind sac, crop, and heart were extracted and stored at −80 °C for later use in RNA and protein extraction experiments. In addition, several testes tissue samples were fixed in 4% paraformaldehyde in phosphate-buffered saline (PBS) for immunofluorescence.

### 3.2. RNA and Protein Extraction and Reverse Transcription

Total RNA and proteins were extracted from tissues using TRIzol Reagent (CWBIO, Taizhou, China) and RIPA Lysis Buffer (Beyotime, Shanghai, China), respectively, according to the manufacturer’s instructions. The PrimeScript^®^ RT Reagent Kit (Takara, Dalian, China) and the SMARTer^TM^ RACE cDNA Amplification Kit (Takara, Dalian, China) were used for reverse transcription to clone the intermediate segment sequence and for 5′ and 3′ RACE reverse transcription, respectively. The mitochondria of *O. tankahkeei* testes in stage IV of spermiogenesis were isolated, and the cytoplasm was removed using the Mitochondria Isolation Kit for Tissue (MedChemExpress, MCE, Monmouth Junction, NJ, USA).

### 3.3. Full-Length Complementary Acquisition and Sequence Analyses of Ot-phbs

The *Ot*-*phb1* and *Ot*-*phb2* full-length complementary DNA (cDNA) sequences were separated according to previously described methods [21]. The primers that were used are listed in Table 1. Various online tools were utilized to predict the protein structure of PHBs, employing previously established methods [22] as outlined in Table 2. The amino acid sequence of *Ot*-PHB1 and *Ot*-PHB2 proteins were aligned with their homologs from other species using Vector NTI10 software. The phylogenetic tree of *Ot*-PHB1, *Ot*-PHB2, and their homologs was constructed by the neighbor-joining method in MEGA 5.1. The GenBank accession numbers of the species selected for multiple sequence alignment and the phylogenetic tree are shown in Appendix A.

### 3.4. Quantitative Analysis of phb mRNA Expression in Different Tissues of O. tankahkeei

Quantitative real-time PCR (qPCR) was used to detect mRNA expression of *phbs* in different tissues (muscle, testis, rectum, hepatopancreas, gill, salivary gland, stomach, kidney, blind sac, crop, and heart) of *O. tankahkeei*. The relative expression levels of *phbs* mRNA were calculated using the cDNA of each tissue as the template for PCR amplification. The amplification efficiency of primers (Table 1) has been verified for qPCR. The amplifications were carried out in a reaction volume of 20 μL containing 5 μL of 1:50 diluted cDNA, 1 μL of each primer (10 μM), 10 μL 2× SYBR Premix Ex Taq II, and 3 μL PCR-grade water. The qPCR was conducted at 95 °C for 10 min, followed by 40 amplification cycles (denaturation at 95 °C for 10 s, annealing at 60 °C for 15 s, and extension at 72 °C for 15 s). Fluorescence readings were recorded at 72 °C after each cycle. According to previous studies [16,23,24], *β-actin* was selected as the reference gene and primers are listed in Table 1 The comparative ΔCt method was used to analyze gene expression. We utilized the independent samples T-test function of SPSS 16.0 software (version 16.0, IBM, Armonk, NY, USA) to determine statistical significance. * denotes *p* < 0.05 and ** denotes *p* < 0.01.

### 3.5. Antibodies

The specific antibodies that were used in this study are listed in this paragraph. Firstly, commercial mouse anti-PHB1(abs100402), and commercial rabbit anti-PHB2 (abs131342) were purchased from Absin Bioscience Inc. (Shanghai, China). SDHA Rabbit pAb (A2594) was obtained from Abclone (WuhanChina). Horseradish peroxidase (HRP)-conjugated goat anti-mouse IgG (H+L) (A0216), anti-rabbit IgG (H+L) (A0208), rabbit anti-polyubiquitin antibody (AF0306), Mito-Tracker Green (C1048), Alexa Fluor 555-labeled goat anti-mouse IgG (H+L) (A0460), and Alexa Fluor 488-labeled goat anti-rabbit IgG (H+L) (A0423) were all purchased from Beyotime (Shanghai, China).

### 3.6. Western Blotting

Western blotting (WB) was used to detect the distribution of *Ot*-PHB1 and *Ot*-PHB2 in mitochondria and mitochondria-free cytoplasm. First, mitochondria were extracted and total cytoplasmic proteins were removed using the Tissue Mitochondria Isolation Kit (C3606; Beyotime, Shanghai, China). After dilution with 5 × sodium-dodecyl sulphate (SDS) buffer, the proteins were boiled for 10 min for structural denaturation. The proteins were then separated on 8% SDS-polyacrylamide electrophoresis gels and transferred to a polyvinylidene difluoride (PVDF) membrane. After blocking with 5% bovine serum albumin for 2 h, the PVDF membrane was incubated with mouse anti-PHB1/rabbit anti-PHB2/SDHA Rabbit pAb (diluted 1:500) at 4 °C overnight. After washing three times with 0.1% PBST (PBS with Tween 20), the membrane was incubated with HRP-conjugated goat anti-mouse IgG (H+L)/HRP-conjugated goat anti-rabbit IgG (Beyotime, Shanghai, China; diluted 1:1500) for 2 h at 37 °C. Then, the membrane was washed five times with 0.1% PBST. Lastly, after the membrane was treated with developer, the emitted signals were visualized using chemiluminescence imaging (Tanon 5200; Shanghai, China). The specificity of commercial mouse anti-PHB1 and commercial rabbit anti-PHB2 was confirmed in the testis of *O. tankahkeei* via WB. The band exhibited singularity and corresponded to the target protein’s size. Therefore, they are suitable for subsequent experiments.

### 3.7. Immunofluorescence

Immunofluorescence was performed as follows. The frozen sections were dried at room temperature for 10 min. The sections were permeabilized in 0.3% PBST (PBS containing 0.3% Triton-X-100) at room temperature for 15 min and blocked with 5% bovine serum protein) for 1.5 h at room temperature. (1:75 dilution); mouse anti-PHB1 antibody and rabbit anti-PHB2 antibody were used as primary antibodies for the expression analysis of PHB1 and PHB2, respectively.

Mouse Anti-PHB1 antibody is the primary antibody for the colocalization of PHB1 with mitochondria. Mouse Anti-PHB1 and rabbit anti-polyubiquitin antibodies are the primary antibodies used for colocalization analysis of PHB1 with polyubiquitin. The sections were washed with PBST three times for 15 min each. The secondary antibody working solution was added to the sections and incubated for at 37 °C 2 h (1:500 dilution). Alexa Fluor 555-labeled goat anti-mouse IgG (H+L) and Alexa Fluor 488-labeled goat anti-rabbit IgG (H+L) were used as secondary antibodies for expression analysis of PHB1 and PHB2, respectively. The secondary antibody for colocalization analysis of PHB1 and mitochondria was Alexa Fluor 555-labeled goat anti-mouse IgG (H+L) supplemented with MitoTracker. The secondary antibodies for collocation analysis of PHB1 and polyubiquitin were Alexa Fluor 555-labeled goat anti-mouse IgG (H+L) and Alexa Fluor 488-labeled goat anti-rabbit IgG (H+L). Tissue sections were stained with 4′,6-diamidino-2-phenylindole (DAPI) for 5 min in the dark. After rinsing with 0.1% PBST, anti-fluorescence quenching was added to the slides. The experimental results were observed and photographed using a laser confocal microscope (LSM880).

### 3.8. RNA Interference

Small interfering RNAs (siRNAs) specific to *phb1* and the negative control (NC) were designed and synthesized by GenePharma (Shanghai, China), and are listed in Table 1. For RNA interference (RNAi), 250 μL phb1 siRNA (approximately 66 μg) and an equal volume of Lipo6000TM transfection reagent were mixed with 2 mL of PBS to prepare the transfection solution. Male *O. tankahkeei* (35–42 g) were randomly divided into two groups, a negative control group and a treated group, with five males in each group. A siRNA mixture of phb1 (500 μL each) was injected into the treated group, and a siRNA mixture of NC (500 μL each) was injected into the control group. At 24 h post-transfection, the injection was repeated. At 48 h post-transfection, control and treated testes were harvested to assess the interfering efficiency and follow-up experiments. qPCR and WB were used to detect the interference efficiency of *phb1*.

### 3.9. Effects of phb1 Knockdown on mtDNA Content

DNA was extracted from the testes of three *O. tankahkeei* using a DNA extraction kit, and the DNA was used as a template for qPCR. mtDNA was amplified by *cytochrome b* (*cytb*) and tubulin was used as the internal reference gene.

### 3.10. Effect of phb1 Knockdown on mRNA Expression of Apoptosis-Related Genes

qPCR was used to detect the effect of *phb1* knockdown on the mRNA expression of *caspase3*, *bcl2*, and *bax*. *β-actin* was used as the reference gene. The comparative ΔCt method was used to analyze gene expression. The primers are shown in Table 1.

### 3.11. Effect of phb1 Knockdown on ROS Level

We measured H_2_O_2_ levels after *phb1* knockdown in the testis by using of the Hydrogen Peroxide Assay Kit (Beyotime, Shanghai, China), and the data were expressed as mean ± standard deviation.

## 4. Discussion

PHBs play a crucial role in maintaining normal mitochondrial functions, and understanding their structure and function is essential for understanding where PHBs are located in the mitochondria. PHBs encompass PHB1 and PHB2, share over 50% similarity and are evolutionarily conserved across all phyla. Both PHBs contain three conserved domains: the C-terminal coiled coil, SPFH, and N-terminal transmembrane domains [2,4]. The basic unit of PHBs formed when PHB1 interacts with PHB2 via the C-terminal coiled-coil domain and the mitochondrial targeting sequence at the N-terminal domains of PHB1 and PHB2 contain a transmembrane region of approximately 20 amino acids each to anchor PHBS to the IMM [4,25]. In this study, the protein structures of *Ot*-PHB1 and *Ot*-PHB2 were predicted and analyzed, and both were found to contain the C-terminal coiled coil, SPFH, and N-terminal transmembrane domains, which are similar to homologous proteins in other species. Further, multiple sequence alignment revealed that these structures were conserved across species, suggesting that their function may also be conserved. In addition, *Ot-*PHB1 and *Ot-*PHB2 were highly colocalized during spermiogenesis, suggesting that they may function primarily as an *Ot*-PHB complex in *O. tankahkeei* spermiogenesis.

*Ot*-*phb1* and *Ot*-*phb2* mRNA were detected in all tissues of *O. tankahkeei*. Their roles in several biological functions, including signal transduction, transcription regulation, cell proliferation, and mitochondrial function may account for their widespread occurrence [26]. Among all tissues, the mRNA expression levels of *Ot*-*phb1* and *Ot*-*phb2* were highest in the testis, which is consistent with previous research indicating that *phb* mRNA is expressed in all tissues, and the expression level is higher in the testis of *Cherax quadricarinatus* [27], *Cynops orientalis* [28], *Boleophthalmus pectinirostris* [17], and *Procambarus clarkia* [29]. These results suggest that *Ot*-PHBs may play an important role in *O. tankahkeei* spermiogenesis.

PHB1 and PHB2 have been shown to exist in the nucleus, mitochondria, and cytoplasm, and are associated with certain cell membrane receptors. Their extensive localization determines their participation in multiple cell biological processes [30]. In the mitochondria, PHB1 and PHB2 form an alternating heterodimeric ring-like complex that is required for mitochondrial stability. In contrast, in the nucleus and cytoplasm, PHB1 and PHB2 act independently of one another [8,31]). Some studies have shown that PHBs are primarily localized in the mitochondria during animal spermiogenesis [17,28,29]. Similarly, in this study, WB showed that *Ot*-PHB1 was mainly expressed in the mitochondria of testes and IF showed that *Ot*-PHB1 and mitochondria were colocalized in the spermiogenesis of *O. tankahkeei*, suggesting that *Ot*-PHBs may play a role in mitochondrial function during spermiogenesis. However, *Ot*-PHB2 was determined to be expressed not only in the mitochondria of testes, but also in cells excluding mitochondria, which may be related to other functions of the PHB2 monomer. Studies have shown that PHB2 is a multifunctional protein located in the IMM, which plays an important role in maintaining the stability of mitochondrial morphology and function and is also an important regulator of cell homeostasis and differentiation. It is directly involved in many cellular processes and plays important roles, such as regulating the activity of transcription factors, regulating cell differentiation and apoptosis, regulating sister chromatid binding, nerve cell repair and regeneration, axon development and formation, and enhancing oxidative stress tolerance of cells [32]. However, whether PHB2 has a similar effect in *O. tankahkeei* remains unclear and requires further investigation.

However, whether PHB2 has a similar effect in *O. tankahkeei* remains unclear and requires further investigation.

PHBs are essential functional proteins that are important for ensuring the normal structure of mitochondria and can be associated with mtDNA to stabilize the mitochondrial genome [6,33,34]). In this study, *phb1* knockdown decreased mtDNA content, indicating that PHBs are critical for maintaining the normal mtDNA content during spermiogenesis in *O. tankahkeei.* This result is in agreement with the reports by Zhang et al. [35] in mice. Studies show that mtDNA content is closely related to spermiogenesis and sperm quality, with poor sperm quality exhibiting increased mtDNA content. May-Panloup et al. [36] reported higher mtDNA content in the sperm of infertile patients and Amaral et al. [37] found that mtDNA content was increased in poor-quality sperm. Hence, we speculate that PHBs play an important role in ensuring sperm quality by maintaining stable mtDNA content. PHBs also act as molecular chaperones, stabilizing mitochondrial complex I protein components [38], and may consequently limit the production of mtROS [39]. In this study, *phb1* knockdown increased the level of mtROS, indicating that PHBs affected production of mtROS during spermiogenesis in *O. tankahkeei.* Sperm produce excessive mtROS, resulting in poor sperm quality. Koppers et al. [40] showed that mitochondrial inhibitors caused mitochondria in sperm to produce excessive ROS, resulting in oxidative damage in the middle part of the sperm cell and loss of sperm motor ability. Therefore, we speculated that PHBs played a significant role in ensuring the quality of sperm by maintaining the stable production of mtROS. Studies have shown that PHBs may inhibit mitochondria-induced apoptosis. Chowdhury et al. [41] found that PHBs had antiapoptotic effects in undifferentiated granulosa cells. Chowdhury et al. [42] demonstrated that PHB1 silencing induced apoptosis in rat granulosa cells. Zhang et al. [35] reported that *phb1* knockout resulted in apoptosis of some spermatocytes. In this study, *phb1* knockout resulted in significantly increased expressions of *caspase-3* and *bax* mRNA in spermatids, indicating that PHBs are involved in regulating apoptosis during spermiogenesis in *O. tankahkeei.*

During spermiogenesis, mitochondria undergo ubiquitination modifications to degrade damaged or excess mitochondria, as observed in *Bos taurus*, *Eriocheir sinensis*, *Procambarus clarkia*, and *Phascolosoma esculenta*, where the colocalization of mitochondrial and polyubiquitin signals has been detected in early to mature spermatids [22,29,43,44]. PHB1 is a substrate for sperm mitochondrial ubiquitination [44]. During the spermiogenesis of *Eriocheir sinensis*, *Charybdis japonica*, and *Procambarus clarkii*, PHB1 colocalized with polyubiquitin [9,29,43]). Therefore, PHB1 may be involved in mitochondrial ubiquitination as a substrate for ubiquitin, and may play a crucial role in the degradation of damaged or excess mitochondria during spermiogenesis [9,29,43,45]. However, it remained unclear whether this ubiquitination mechanism existed in cephalopods. In this study, *Ot*-PHB1 colocalization with polyubiquitin was detected throughout *O. tankahkeei* spermiogenesis. Notably, the signal patterns of ubiquitin-PHB1 colocalization and PHB1-mitochondrial colocalization were extremely similar and synchronous, confirming that PHBs play a key role in spermatogenesis in cephalopods, possibly by mediating mitochondrial ubiquitination.

## 5. Conclusions

In this study, we found that the sequence and distribution of *Ot*-PHB1 and *Ot*-PHB2 were conserved and highly colocalized; thus, we inferred that their functions were conserved and that they may function primarily as an *Ot*-PHB complex. *Ot-*PHB1 proteins were mainly expressed and localized in mitochondria during the spermiogenesis of *O. tankahkeei*. In addition, *Ot-*PHB1 was colocalized with polyubiquitin during spermiogenesis, suggesting that *Ot-*PHB1 may be a substrate of polyubiquitin that regulates the ubiquitination of mitochondria to ensure mitochondrial quality during spermiogenesis of *O. tankahkeei*. To further investigate the function of PHBs in the spermiogenesis of *O. tankahkeei*, we knocked down *phb1* and found that *Ot-*PHBs may affect mitochondrial function by maintaining mtDNA content and stabilizing ROS levels. In addition, they may affect spermatocyte survival by regulating mitochondria-induced apoptosis in the spermiogenesis of *O. tankahkeei*. These results provide evidence for the hypothesis that PHBs are involved in spermiogenesis by influencing mitochondrial function, and they establish a foundation for further research into the molecular mechanisms of spermatogenesis in *O. tankahkeei* and other cephalopods.

## Figures and Tables

**Figure 1 ijms-24-10030-f001:**
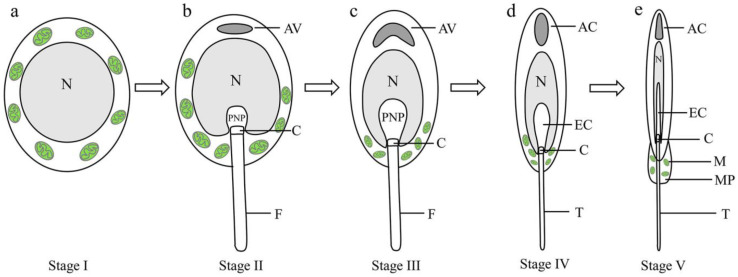
Dynamic changes of mitochondria during the stages of spermiogenesis of *O. tankahkeei.* (**a**) Stage I. (**b**) Stage II. (**c**) Stage III. (**d**) Stage IV. (**e**) Stage V. N: nucleus; M: mitochondria; AV: acrosomal vesicle; PNP, posterior nuclear pocket; C: centrioles; F: flagellum; AC: acrosomal cone; EC: endonuclear channel; MP: midpiece; T: Tail.

**Figure 2 ijms-24-10030-f002:**
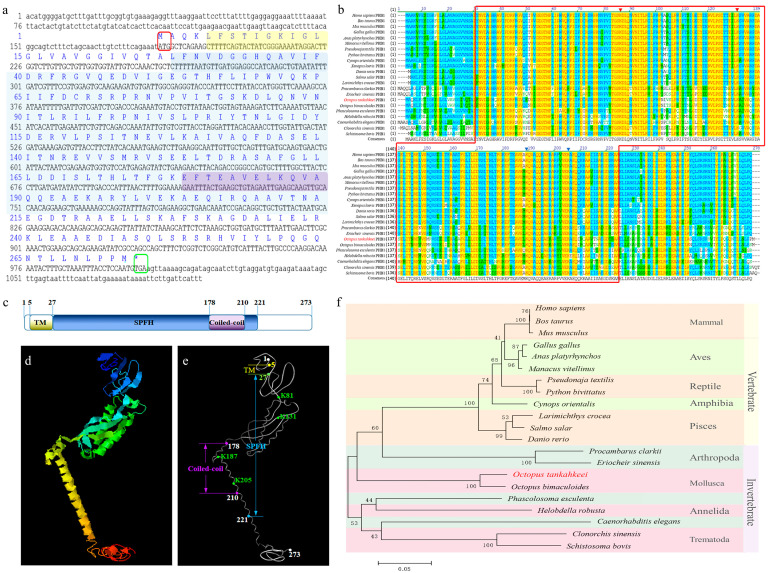
Analysis of *Ot-phb1* gene characteristics. (**a**) Full-length cDNA of *Ot*-*phb1* with the deduced amino acid sequence. Lowercase letters denote the 5′ and 3′ UTR, respectively. The blue letters above the base sequence represent the corresponding amino acid sequence. The yellow, blue, and purple shading represent the possible transmembrane, SPFH, and coiled-coil domains of the PHB1 protein, respectively. The red boxes represent the start codons and the green boxes represent stop codons. (**b**) Multiple sequence alignment of *Ot*-PHB1 proteins with homologs from other species. (**c**) The protein domains of *Ot*-PHB1. The red and blue triangles indicate the conserved lysine sites in *Ot*-PHB1, where the blue triangles indicate lysine sites in the coiled-coil domain. (**d**,**e**) The predicted tertiary structure of *Ot*-PHB1. TM: transmembrane domain; Coiled-coil: coiled-coil domain; SPFH: SPFH domain; K: the conserved lysine site. (**f**) Phylogenetic tree based on the amino acid sequence of PHB1.

**Figure 3 ijms-24-10030-f003:**
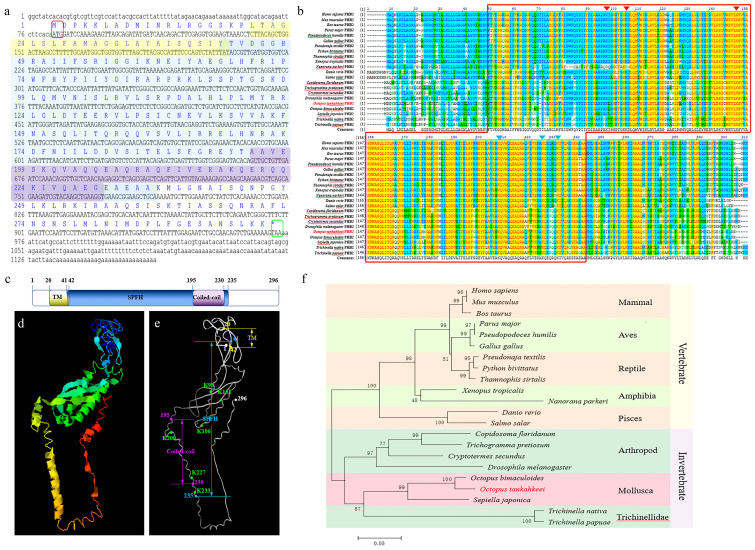
Analysis of *Ot-phb2* gene characteristics. (**a**) Full-length cDNA of *Ot*-*phb2* with the deduced amino acid sequence. Lowercase letters denote the 5′ and 3′ UTR, respectively. The blue letters above the base sequence represent the corresponding amino acid sequence. The yellow, blue, and purple shading represent the possible transmembrane, SPFH, and coiled-coil domains of the PHB2 protein, respectively. The red boxes represent the start codons and the green boxes represent stop codons. (**b**) Multiple sequence alignment of *Ot*-PHB2 proteins with homologs of other species. The red and blue triangles indicate the conserved lysine sites in the *Ot*-PHB2, where the blue triangles indicate lysine sites in the coiled-coil domain. (**c**) The protein domains of *Ot*-PHB2. (**d**,**e**) The predicted tertiary structure of *Ot*-PHB2. TM: transmembrane domain; Coiled-coil: coiled-coil domain; SPFH: SPFH domain. K: the conserved lysine site. (**f**) Phylogenetic tree based on the amino acid sequence of PHB2.

**Figure 4 ijms-24-10030-f004:**
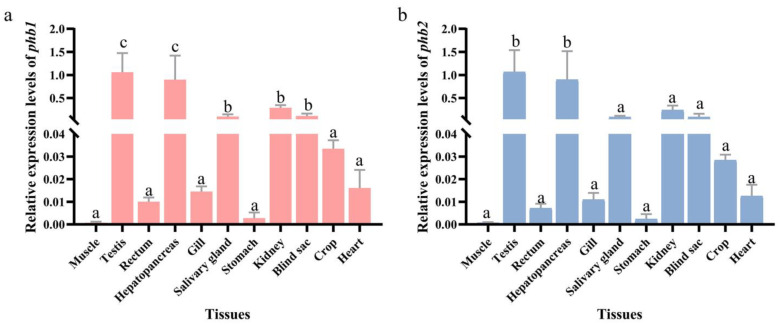
Tissue expression characteristics of PHBs in *O. Tankahkeei.* (**a**) Expression characteristics of *Ot*-*phb1* in different tissues. (**b**) The expression characteristics of *Ot*-*phb2* in different tissues. Different letters indicate significant differences between groups, *p* < 0.05.

**Figure 5 ijms-24-10030-f005:**
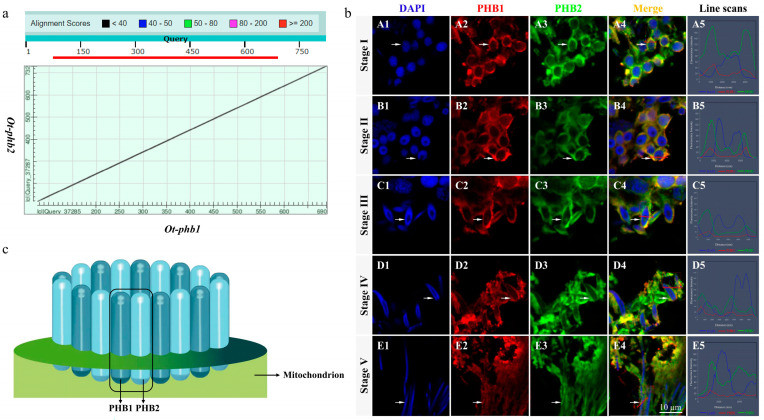
*Ot*-PHB1 and *Ot*-PHB2 function as a PHB complex during spermiogenesis in *O. Tankahkeei*. (**a**) Sequence alignment of *Ot*-PHB1 and *Ot*-PHB2 proteins (dot matrix) revealed that *Ot*-PHB1 and *Ot*-PHB2 proteins were highly homologous subunits with approximately 54.15% amino acid sequence identity (**b**) Expression and colocalization of PHB1 and PHB2 proteins in the spermiogenesis of *O. tankahkeei*. (A1–A5) Stage I of spermiogenesis. (B1–B5) Stage II of spermiogenesis. (C1–C5) Stage III of spermiogenesis; (D1–D5) Stage IV of spermiogenesis. (E1–E5) Stage V of spermiogenesis; blue, red, and green denote the nucleus stained with DAPI, PHB1 protein, and PHB2 protein, respectively. (**c**) The mode pattern of the PHB complex. Prohibitins (PHBs) comprise 12 to 16 PHB1-PHB2 units that predominantly localize in the inner mitochondrial membrane.

**Figure 6 ijms-24-10030-f006:**
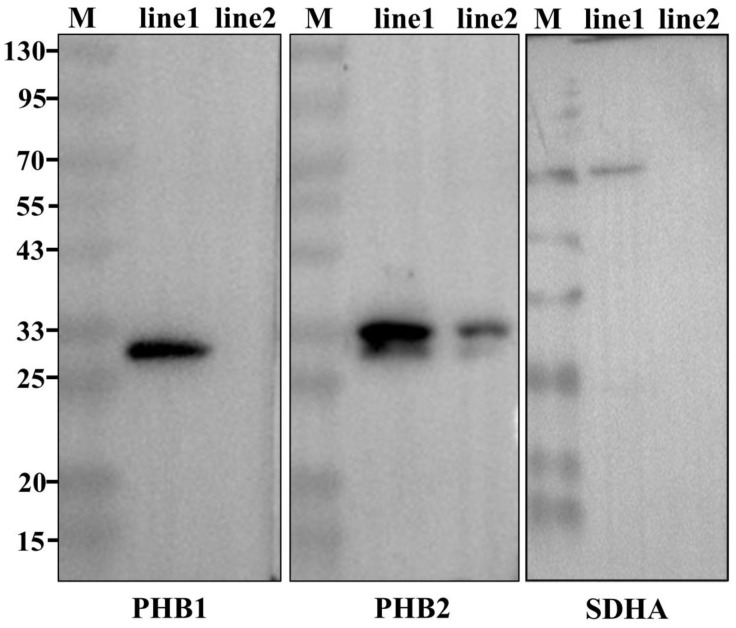
Expression analysis of PHB1 and PHB2 in mitochondria. Succinate dehydrogenase complex, subunit A (SDHA) is the indicator of mitochondria. M: Protein marker; line1: total mitochondrial protein of testis; line2: total cytoplasmic protein of testes after removal of mitochondria.

**Figure 7 ijms-24-10030-f007:**
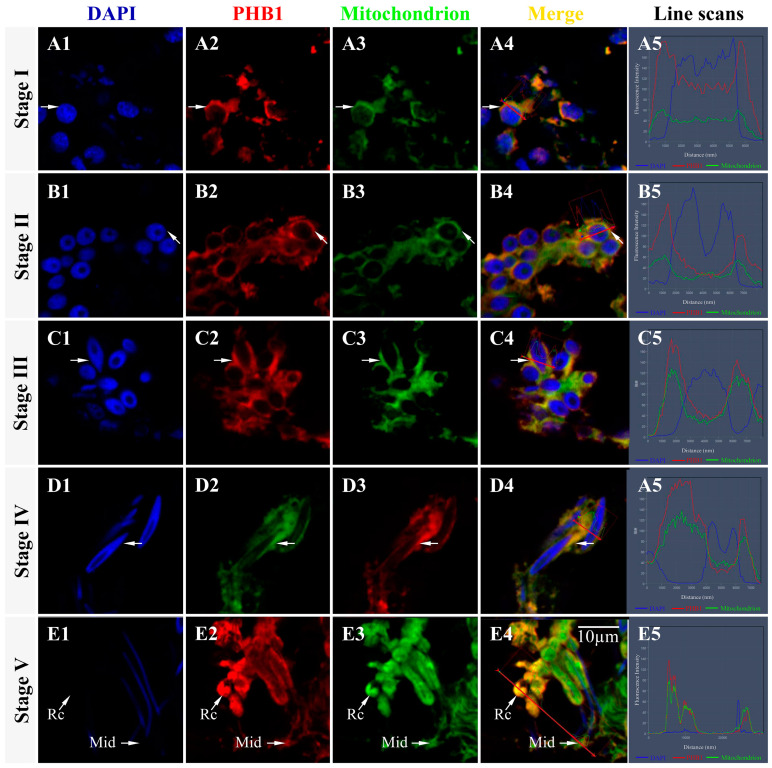
Colocalization of the PHB1 protein and mitochondria in the spermiogenesis of *O. tankahkeei*. (**A1**–**A5**) Stage I of spermiogenesis. (**B1**–**B5**) Stage II of spermiogenesis. (**C1**–**C5**) Stage III of spermiogenesis. (**D1**–**D5**) Stage IV of spermiogenesis. (**E1**–**E5**) Stage V of spermiogenesis. Mid: midpiece. Rc: residual cytoplasm. Blue, red, and green denote the nucleus stained with DAPI, PHB1 protein, and mitochondrion, respectively.

**Figure 8 ijms-24-10030-f008:**
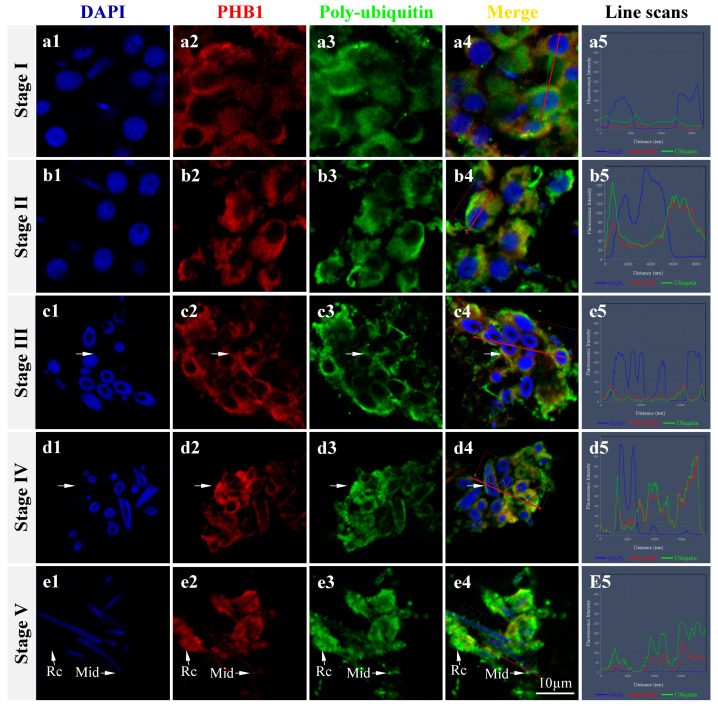
Colocalization of PHB1 protein and polyubiquitin in the spermiogenesis of *O. tankahkeei.* (**A1**–**A5**) Stage I of spermiogenesis. (**B1**–**B5**) Stage II of spermiogenesis. (**C1**–**C5**) Stage III of spermiogenesis. (**D1**–**D5**) Stage IV of spermiogenesis. (**E1**–**E5**) Stage V of spermiogenesis. Mid: midpiece. Rc: residual cytoplasm. Blue, red, and green denote the nucleus stained with DAPI, PHB1 protein, and polyubiquitin, respectively.

**Figure 9 ijms-24-10030-f009:**
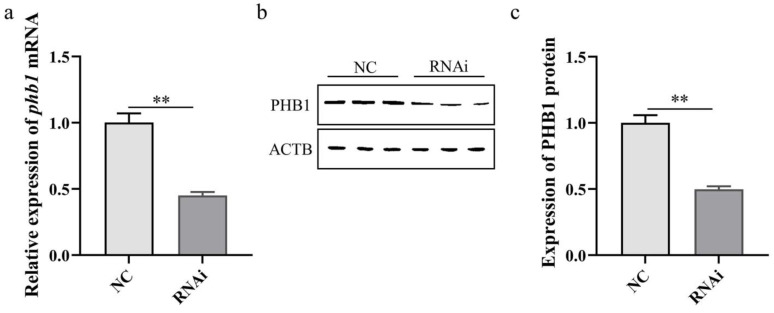
Detection of *phb1* RNAi efficiency. (**a**) Effect of *phb1* knockdown on the expression of *phb1* mRNA. (**b**,**c**) Effect of *phb1* knockdown on the expression of PHB1 protein. NC: control group; RNAi: experimental group. ** represents *p* < 0.01.

**Figure 10 ijms-24-10030-f010:**
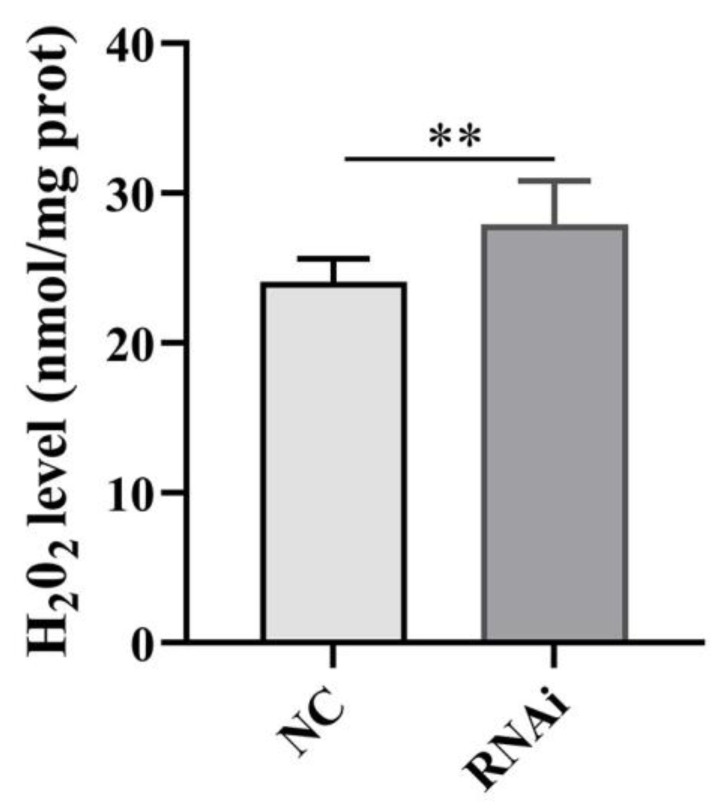
Effect of phb1 knockdown on ROS levels. NC: negative control group; RNAi: experimental group. ** represents *p* < 0.01.

**Figure 11 ijms-24-10030-f011:**
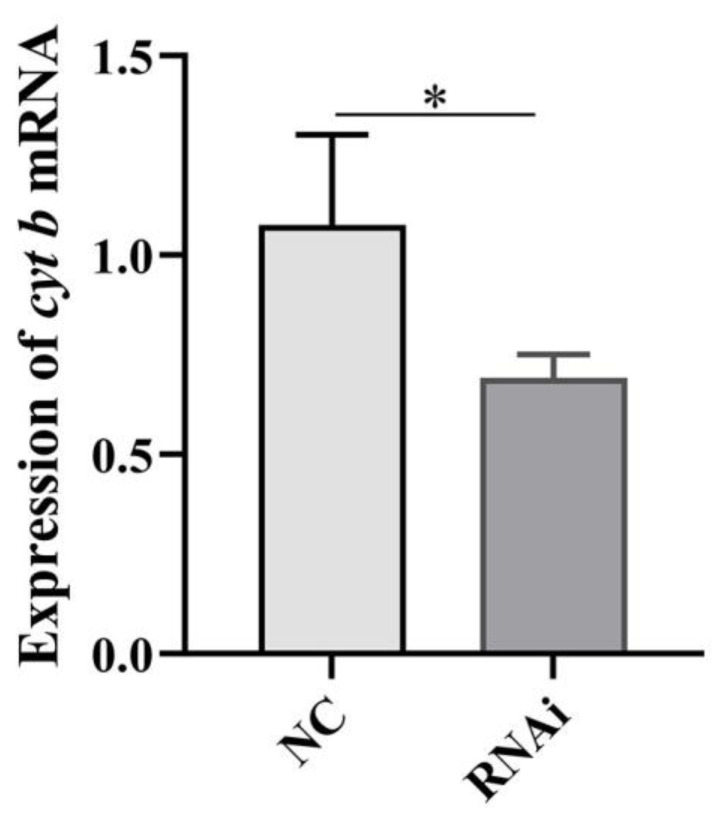
Effect of *phb1* knockdown on copy numbers of mtDNA. NC: negative control group; RNAi: experimental group. * denotes *p* < 0.05.

**Figure 12 ijms-24-10030-f012:**
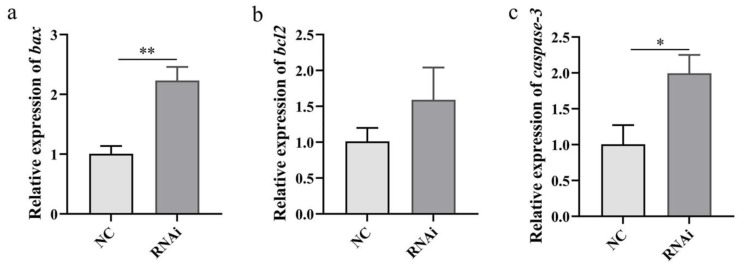
Effect of *phb1* knockdown on *bax* (**a**), *bcl2* (**b**), and *caspase-3* (**c**) mRNA expression. NC: negative control group; RNAi: experimental group. * denotes *p* < 0.05; ** represent *p* < 0.01.

**Table 1 ijms-24-10030-t001:** Sequences of the primers that were used in the study.

Primer	Sequence (5′–3′)	PCR Product Sizes	Purpose
*phb1*-F	AAGAAGATGTGATTGGCG	660	PCR
*phb1*-R	TATTTTGTCCTTGGGGCA		PCR
*phb2*-F1	GCTTACGCCATTTCCCAGTC	562	PCR
*phb2*-R1	GTTCTTGCTTGGCTCTTTCTACA		PCR
*phb2*-F2	TCAGTGTGCTTATCCGACGAG	263	PCR
*phb2*-R2	TCCAGGGTTTTGTGAGATAGCA		PCR
*phb1*,5′RACE-R	CAACAAGACCAAGTCCTATTTTCCCG	>233	5′ RACE
*phb1*,3′RACE-F1	TGAACTTCGCAAACTGGA	>115	3′ RACE
*phb2*,5′RACE-R1	CACCCGCTCTTCATAATCTAATCCCA	>396	5′ RACE
*phb2*,5′RACE R2	CAATCGTCGGTACATGAGAGGCAG	>372	5′ RACE
*phb2*,3′RACE F1	CGACGATTGGGATTAGATTATGAAGAGC	>528	3′ RACE
*phb2*,3′RACE F2	CGGGTGCTACCATCAATTTGTAACG	>501	3′ RACE
*phb1*-qPCR-F	GCTTTTGGCTTACTCCTTGA	151	qPCR
*phb1*-qPCR-R	CAGCCTGTCGGATTTGTTC		qPCR
*phb2*-qPCR-F	AGAAGGGCTACATTTCAGGA	185	qPCR
*phb2*-qPCR-R	CTAATCCCAATCGTCGGTA		qPCR
*β-actin*-F	CCCATCTATGAAGGTTACGC	161	qPCR
*β-actin*-R	GAGATTCTGGGCACCTAAAC		qPCR
*cytb*-F	ATTTTGAGGGGCTACGG	156	semi-quantitative PCR
*cytb*-R	CCCACAATGATAAACGGAA		semi-quantitative PCR
*tubulin*-F	TTTCTCCCCGAGCCAATG	181	semi-quantitative PCR
*tubulin*-R	GGATACCGCCACAAACCTGAC		semi-quantitative PCR
siRNA-*phb1*-F	GAGGCCAUCAAGCUGUAAUTT	-	RNA interference
siRNA-*phb1*-R	AUUACAGCUUGAUGGCCUCTT		RNA interference
siRNA-NC-F	UUCUCCGAACGUGUCAGGUTT	-	Negative siRNA
siRNA-NC-R	ACGUGACACGUUCGGAGAATT		Negative siRNA

**Table 2 ijms-24-10030-t002:** Protein structure online analysis tool.

Forecasting Tool	Function	Website
The Sequence Manipulation Suite	Analysis and formatting of DNA and protein sequences	http://www.bio-soft.net/sms/; accessed on 1 June 2023
COILS	Prediction of coiled-coil regions in proteins	http://www.ch.embnet.org/software/COILS_form.html; accessed on 1 June 2023
ExPASy ProtParam	Prediction of molecular weight and isoelectric point in proteins	https://www.expasy.org/; accessed on 1 June 2023
TMHMM Server v.2.0	Prediction the structure of transmembrane domains in proteins	http://www.cbs.dtu.dk/services/TMHMM/; accessed on 1 June 2023
The NCBI CD-search tool	Prediction of protein domains	https://blast.ncbi.nlm.nih.gov/Blast.cgi; accessed on 1 June 2023
TASSER	Prediction of the tertiary structure in proteins	https://zhanglab.ccmb.med.umich.edu/I-TASSER/; accessed on 1 June 2023

## Data Availability

The authors confirm that the data supporting the findings of this study are available within the article and its Appendix A.

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
