# Peer review of "The Effect of Prohibitins on Mitochondrial Function during Octopus tankahkeei Spermiogenesis"

_ijms, 2023, doi:10.3390/ijms241210030_

Round 1

Reviewer 1 Report

The study provides data about prohibitins 1 and 2 in spermiogenesis of a cephalopod species Octopus tankahkeei. The work can be possibly beneficial for establishment of aquaculture and thus for conservation of the species in its natural habitat. To attain this goal, the manuscript should be thoroughly revised and additional experiments performed. Then the manuscript should be reconsidered for publication in International Journal of Molecular Sciences. The comments can be found below:

Comments towards methods

Section 2.3

This section need to be completely rewritten. The current structure (list of tools then list of the tasks performed) is confusing. Maybe a table (in the main text or in the supplement) can help. Also, although the tools are available online for everybody, they were usually described formally in scientific publications and these publications should be cited properly. Information about the appropriate references can be found on the web pages dedicated to the tools.

Sections 2.4 and 2.8

Semiquantitative PCR should be replaced by quantitative real-time or digital PCR. The quantitative PCR is available to the authors and they use it here. Why they use the semiquantitative method at all? It is more laborious, more prone to human error, and the results are much less reproducible. Further, the selection of the endogenous control gene should be specified in more detail.

Section 2.5 and 2.6

Mouse anti-PHB1 and rabbit anti-PHB2 antibodies should be clearly specified. The reference (Wang et al. 2022) provided in section 2.5 does not seem to discuss these particular antibodies (antibody against Kif17 is tested there, and there is nothing beyond standard Western blotting procedure in the methods of that paper.). Only manufacturer is revealed in section 2.6, there is no information about catalogue number and it is not known whether the antibodies were obtained by immunization with Ot PHB proteins or PHBs from a different species - in that case it would be helpful to specify why the authors thought that the antibodies will show specific cross reactivity against Ot PHBs and evidence should be presented for the cross-reactivity. It is absolutely required to do so for the work to be reproducible. Further, I would recommend to confirm the specificity of the antibody by blocking its binding with a corresponding synthetic peptide or recombinant protein prior to Western blotting.

Section 2.7

“For RNA interference, 250 μL phb1 siRNA and an equal volume of Lipo6000TM regent…” (section 2.7)

Volume is insufficient, concentration has to be also specified otherwise the experiment cannot be reproduced.

Comments for Results

section 3.1

“tertiary structures were predicted using online tools (Fig. 2d, f)”.

The corresponding parts of figure 2 should be 2d,e.

“Ot-PHB1 was found to have a close evolutionary relationship with PHB1 in Octopus bimaculoides, which also belongs to Cephalopoda.”

Why the % identity is not listed for comparison of PHBs between Ot and Ob, and why Ob is not present in Fig2f?. Description of panel f is missing in legend of figure 2.

Section 3.2

Figure 3 is identical to figure 2, Its legend contain the same errors as Fig2 legend.

For sections 3.1 and 3,2, giving the % protein identity in directly in the text is rather confusing. I suggest that the Supplementary tables 1 and 2 should contain the % sequence identity for all species, the main text in results section can then only contain the range.

Section 3.4.

“Ot-PHB1 and Ot-PHB2 function as a PHB complex during O. tankahkeei spermiogenesis”

This is surely an overstatement, since there is only partial colocalization as revealed by diffraction limited confocal microscope. This is similar for section 3.5 and 3.6.

It is true, that fro mitochondrial localization there is also biochemical evidence - although I would like to see more fractions in addition to mitochondria. Biochemical analysis (fig.6) also suggests that while PHB1 is restricted to mitochondria, PHB2 is present also in cytoplasm (and who knows where else), and therefore may function partially independently of the presumable PHB1-PHB2 complex. From my point of view, there is only a weak evidence of colocalization without further biochemical or detailed microscopic imaging support. I need to stress that I have insufficient hands on expertise in confocal microscopy, but I think that in all figures 5,7 and 8 there is basically a cytosolic signal which is rather nonspecific, I believe we would see the same level of "colocalization" with e.g. glycolytic enzymes. I would suggest e.g. coimmunoprecipitation experiments to confirm independently a functional relationship between PHN1 and PHB2 and between PHBs and the mitochondria polyubiquitination.

Figures 5,7,8 - Line Scans legend is illegible - I suggest the authors provide larger panels in supplementary data. Furthermore, there are several evident errors, e.g. the merged images are often inverted compared to the individual channel images, and in fig.7 d2 and d3 colors or panels are swapped.

Section 3.7-10 and figure 9-12.

RNAi was in my opinion not very effective. The RNA/protein reduction is statistically significant, but modest - protein level is approximately at 60% compared to wild-type. Consequently the phenotype effects are also quite small, e.g. I do not believe that mtDNA content presented in figure 11 is really significantly lower in RNAi compared to wild type - by naked eye I see the difference only for the third RNAi sample and the error bars significantly overlap (by the way, which quantity is represented by the error bars?). Moreover, how to explain that in infertile patients, the tendency is for increased mtDNA content? The authors surely cannot say that bcl-2 expression is increased, since it is not statistically significant. Further, the presumable proapoptotic effect of PHB1 knockdown would be discrepant with increase of antiapoptotic bcl-2 (it could be a negative feedback, nothing more). I require here revision of the statistical tests to ascertain that the significance is duly assigned. Furthermore,  what I miss here is any information about the state of the sperm. Was it morphologically or functionally abnormal after RNAi or was there no difference between the groups?

I am not a native speaker. My comments need to be taken with caution. However, I do see multiple instances where the English is not sufficiently clear and easy to read. So I recommend a professional editing service. Below I give only few examples (there are more to correct than what is listed below!)

"Further, Ot-PHB1 and Ot-PHB2 were highly colocalized, suggesting they may function primarily as Ot-PHBs in O. tankahkeei. Ot-PHB1 proteins were mainly expressed and localized in mitochondria during spermiogenesis, implying that they may function in mitochondria" (lines 18-21)

These two sentences do not have any meaning. If I can be allowed an analogy, this is like saying I saw the rat 1 and the rat 2 together, so they probably function as rats. I saw them in a sewage tube, so they probably live in the sewage tube.

"colocated" (line 21) colocalized

(mitochondria) "store energy for sperm motility" (line 54) provide

"studying the sperm shape of O. tankahkeei can provide a theoretical basis for artificial breeding" (line 73-4) "shape" should be omitted

"lookup SPFH domain of PHBs I" (lines 115-6) unintelligible

Author Response

Reviewer 2

Comments and Suggestions for Authors

The study provides data about prohibitins 1 and 2 in spermiogenesis of a cephalopod species Octopus tankahkeei. The work can be possibly beneficial for establishment of aquaculture and thus for conservation of the species in its natural habitat. To attain this goal, the manuscript should be thoroughly revised and additional experiments performed. Then the manuscript should be reconsidered for publication in International Journal of Molecular Sciences. The comments can be found below:

Comments towards methods

Section 2.3

This section need to be completely rewritten. The current structure (list of tools then list of the tasks performed) is confusing. Maybe a table (in the main text or in the supplement) can help. Also, although the tools are available online for everybody, they were usually described formally in scientific publications and these publications should be cited properly. Information about the appropriate references can be found on the web pages dedicated to the tools.

Answer: Thank you for your valuable suggestion. We have revised section 2.3.

Sections 2.4 and 2.8

Semiquantitative PCR should be replaced by quantitative real-time or digital PCR. The quantitative PCR is available to the authors and they use it here. Why they use the semiquantitative method at all? It is more laborious, more prone to human error, and the results are much less reproducible. Further, the selection of the endogenous control gene should be specified in more detail.

Answer: Thank you for this valuable suggestion. (1) We have eliminated semi-quantitative PCR from 2.4 and incorporated statistical significance across tissues into the qPCR results in 2.4. (2) We concur with your suggestion and have conducted qPCR to re-evaluate the alterations in cyt b (a mitochondrial gene) expression (3) We went through a lot of literature and found that the β-actin gene was often used as an internal control in reproductive biology research (Qin et al., 2018; Jia et al., 2018; Ni et al., 2019 etc). In addition, Wang et al. (2010), Mao et al. (2012) and Long et al. (2015) used β-actin as a reference gene for the study of gonad-development-related genes expression in Octopus tankahkeei. Therefore, we choose β-actin as a internal control. This is a great suggestion. In future studies, we will consider this recommendation and explore whether β-actin can be used as an reference gene for the study of gonad-development-related genes expression.

Qin G. Luo W. Tan S. et al. 2018. Dimorphism of sex and gonad-development-related genes in male and female lined seahorse, Hippocampus erectus, based on transcriptome analyses. Genomics, https://doi:10.1016/j.ygeno.2018.11.008. 

Jia Y. Zheng J. Chi M. et al. 2018. Molecular identification of dmrt1 and its promoter CpG methylation in correlation with gene expression during gonad development in Culter alburnus. Fish Physiology and Biochemistry, https://doi:10.1007/s10695-018-0558-1 

Ni F. Yu H. Liu Y. et al. 2019. Roles of piwil1 gene in gonad development and gametogenesis in Japanese flounder, Paralichthys olivaceus. Gene.  https://doi:10.1016/j.gene.2019.03.045. ),

Wang et al (2010), Mao et al (2012) and Long et al (2015) used β-actin as an reference gene for the study of gonad-development-related genes expression in Octopus tankahkeei.

Long LL, Han YL, Sheng Z, Du C, Wang YF, Zhu JQ. Expression analysis of HSP70 in the testis of Octopus tankahkeei under thermal stress. Comp Biochem Physiol A Mol Integr Physiol. 2015 Sep;187:150-9. doi: 10.1016/j.cbpa.2015.05.022. Epub 2015 May 30. PMID: 26033497.

Mao HT, Wang DH, Lan Z, Zhou H, Yang WX. Gene expression profiles of prohibitin in testes of Octopus tankahkeei (ot-phb) revealing its possible role during spermiogenesis. Mol Biol Rep. 2012 May;39(5):5519-28. doi: 10.1007/s11033-011-1355-4. Epub 2011 Dec 21. PMID: 22187346.

Wang W, Zhu JQ, Yang WX. Molecular cloning and characterization of KIFC1-like kinesin gene (ot-kifc1) from Octopus tankahkeei. Comp Biochem Physiol B Biochem Mol Biol. 2010 Jul;156(3):174-82. doi: 10.1016/j.cbpb.2010.03.004. Epub 2010 Mar 19. PMID: 20304088.

Section 2.5 and 2.6

Mouse anti-PHB1 and rabbit anti-PHB2 antibodies should be clearly specified. The reference (Wang et al. 2022) provided in section 2.5 does not seem to discuss these particular antibodies (antibody against Kif17 is tested there, and there is nothing beyond standard Western blotting procedure in the methods of that paper.). Only manufacturer is revealed in section 2.6, there is no information about catalogue number and it is not known whether the antibodies were obtained by immunization with Ot PHB proteins or PHBs from a different species - in that case it would be helpful to specify why the authors thought that the antibodies will show specific cross reactivity against Ot PHBs and evidence should be presented for the cross-reactivity. It is absolutely required to do so for the work to be reproducible. Further, I would recommend to confirm the specificity of the antibody by blocking its binding with a corresponding synthetic peptide or recombinant protein prior to Western blotting.

Answer: Thank you for this valuable suggestion. Mouse anti-PHB1 and rabbit anti-PHB2 antibodies were commercially purchased (item numbers provided in section 2.5) and are described as available for WB and IF in the product description. We concur with your suggestion. However, as the antibodies in question are commercial, we have not obtained a corresponding synthetic peptide or recombinant protein to validate their specificity through blocking. We aspire to confirm the antibody's specificity (by generating our own) via blocking its binding with an appropriate synthetic peptide or recombinant protein prior to Western blotting in future. In this study, following the acquisition of antibodies, we initially confirmed their specificity in the testis of Octopus tankahkeei via western blotting. The band exhibited singularity and corresponded to the target protein's size. Consequently, we concluded that they are suitable for subsequent experimentation.

Section 2.7

“For RNA interference, 250 μL phb1 siRNA and an equal volume of Lipo6000TM regent…” (section 2.7)

Volume is insufficient, concentration has to be also specified otherwise the experiment cannot be reproduced.

Answer: Thank you for this valuable suggestion. The concentration of siRNA has been included in section 2.7.

Comments for Results

section 3.1

“tertiary structures were predicted using online tools (Fig. 2d, f)”.

The corresponding parts of figure 2 should be 2d,e.

 Answer: We have We sincerely apologize for our error and have corrected it.

“Ot-PHB1 was found to have a close evolutionary relationship with PHB1 in Octopus bimaculoides, which also belongs to Cephalopoda.”

Why the % identity is not listed for comparison of PHBs between Ot and Ob, and why Ob is not present in Fig2f?. Description of panel f is missing in legend of figure 2.

Answer: Thank you for this valuable suggestion. We have revised the multiple sequence alignment diagram in accordance with your suggestion and incorporated a description of panel f into the legend of figure 2

Section 3.2

Figure 3 is identical to figure 2, Its legend contain the same errors as Fig2 legend.

Answer: We have made modifications in accordance with your suggestion.

For sections 3.1 and 3,2, giving the % protein identity in directly in the text is rather confusing. I suggest that the Supplementary tables 1 and 2 should contain the % sequence identity for all species, the main text in results section can then only contain the range.

 Answer: Thank you for this valuable suggestion. We have made modifications in accordance with your suggestion.

Section 3.4.

“Ot-PHB1 and Ot-PHB2 function as a PHB complex during O. tankahkeei spermiogenesis”

This is surely an overstatement, since there is only partial colocalization as revealed by diffraction limited confocal microscope. This is similar for section 3.5 and 3.6.

It is true, that fro mitochondrial localization there is also biochemical evidence - although I would like to see more fractions in addition to mitochondria. Biochemical analysis (fig.6) also suggests that while PHB1 is restricted to mitochondria, PHB2 is present also in cytoplasm (and who knows where else), and therefore may function partially independently of the presumable PHB1-PHB2 complex. From my point of view, there is only a weak evidence of colocalization without further biochemical or detailed microscopic imaging support. I need to stress that I have insufficient hands on expertise in confocal microscopy, but I think that in all figures 5,7 and 8 there is basically a cytosolic signal which is rather nonspecific, I believe we would see the same level of "colocalization" with e.g. glycolytic enzymes. I would suggest e.g. coimmunoprecipitation experiments to confirm independently a functional relationship between PHN1 and PHB2 and between PHBs and the mitochondria polyubiquitination.

Answer: Your suggestions are highly professional and merit our consideration. Research has demonstrated that both PHB1 and PHB2 are present in the nucleus, mitochondria, and cytoplasm, where they interact with specific cell membrane receptors. Their extensive localization enables their involvement in numerous cellular processes (Thuaud et al., 2013). In the mitochondria, PHB1 and PHB2 form a complex of PHBs that is essential for maintaining mitochondrial stability. Conversely, in the nucleus and cytoplasm, PHB1 and PHB2 function independently of each other. (Mishra et al. 2006; Merkwirth et al. 2008). Several studies have demonstrated that PHBs are primarily localized within the mitochondria and play a crucial role in animal spermiogenesis (Dong et al. 2015; Jin et al. 2016; Wang et al. 2017). The primary objective of this investigation is to examine the impact of PHBs on mitochondrial function during O. tankahkeei spermiogenesis. Therefore, we did not further explore the rationale behind the detection of PHB2 in cells devoid of mitochondria. However, we fully concur with your viewpoint that it is inappropriate to infer the involvement of PHB1 and PHB2 in mitochondrial function as complex phbs during O. tankahkeei spermiogenesis solely based on the current findings. This is not methodologically sound, and we will revise the definitive terminology employed in this section of the article. However, in the forthcoming study, we will produce our own antibodies and employ coimmunoprecipitation to validate that PHB1 and PHB2 modulate mitochondrial function as a complex of PHBs. Additionally, we will futher investigate the rationale behind the localization of PHB2 in both cytoplasmic and nuclear compartments. We can generate truncated and amino acid mutants of the PHB2 protein sequence, followed by laser confocal microscopy to observe their expression and localization. This will enable us to identify the specific sequence responsible for PHB2's migration and localization to mitochondria.

Thuaud F, Ribeiro N, Nebigil CG, Désaubry L (2013) Prohibitin ligands in cell death and survival: mode of action and therapeutic potential. Chemistry & biology 20(3):316-31. https://doi.org/10.1016/j.chembiol.2013.02.006

Mishra S, Murphy LC, Murphy LJ (2006) The Prohibitins: emerging roles in diverse functions. Journal of Cellular & Molecular Medicine, 2010, 10(2):353-363. https://doi.org/10.1111/j.1582-4934.2006.tb00404.x

Merkwirth C, Dargazanli S, Tatsuta T, Dargazanli S, Tatsuta T, Geimer S, Löwer B, Wunderlich FT, Kleist-Retzow JC, Waisman A, Westermann B, Langer T (2008) Prohibitins control cell proliferation and apoptosis by regulating OPA1-dependent cristae morphogenesis in mitochondria. Genes & Development 22(4):476-488. https://doi.org/10.1101/gad.460708

Dong WL, Hou CC, Yang WX (2015) Mitochondrial prohibitin and its ubiquitination during crayfish Procambarus clarkii spermiogenesis. Cell and Tissue Research 359(2):679-692. https://doi.org/10.1007/s00441-014-2044-0

Jin JM, Hou CC, Tan FQ, Yang WX (2016) The potential function of prohibitin during spermatogenesis in Chinese fire-bellied newt Cynops orientalis. Cell and Tissue Research 363(3):805-822. https://doi.org/10.1007/s00441-015-2280-y

Wang D, Zhao YQ, Han YL, Hou CC, Zhu JQ (2017) Characterization of mitochondrial prohibitin fromBoleophthalmus pectinirostris and evaluation of its possible role in spermatogenesis. Fish Physiology and Biochemistry 43(5):1299-1313. https://doi.org/10.1007/s10695-017-0373-0

Figures 5,7,8 - Line Scans legend is illegible - I suggest the authors provide larger panels in supplementary data. Furthermore, there are several evident errors, e.g. the merged images are often inverted compared to the individual channel images, and in fig.7 d2 and d3 colors or panels are swapped.

 Answer: Thank you for this valuable suggestion. We will provide larger panels in the supplementary data and have corrected the error depicted in Figure 7.

Section 3.7-10 and figure 9-12.

RNAi was in my opinion not very effective. The RNA/protein reduction is statistically significant, but modest - protein level is approximately at 60% compared to wild-type. Consequently the phenotype effects are also quite small, e.g. I do not believe that mtDNA content presented in figure 11 is really significantly lower in RNAi compared to wild type - by naked eye I see the difference only for the third RNAi sample and the error bars significantly overlap (by the way, which quantity is represented by the error bars?). Moreover, how to explain that in infertile patients, the tendency is for increased mtDNA content? The authors surely cannot say that bcl-2 expression is increased, since it is not statistically significant. Further, the presumable proapoptotic effect of PHB1 knockdown would be discrepant with increase of antiapoptotic bcl-2 (it could be a negative feedback, nothing more). I require here revision of the statistical tests to ascertain that the significance is duly assigned. Furthermore,  what I miss here is any information about the state of the sperm. Was it morphologically or functionally abnormal after RNAi or was there no difference between the groups?

Answer: 1. In this study, the mRNA levels of phb1 were significantly reduced following the knockdown of phb1. During qPCR verification, we conducted three parallel experiments on O. tankahkeei individuals and repeated each experiment three times to minimize individual differences and ensure reliable data. Therefore, we hypothesize that the interference efficiency of protein levels is less than 50% due to individual differences. Following your suggestion, we have re-verified the protein levels and this time selected three individuals for validation.

  1. In this study, the knockdown of phb1 resulted in a reduction of mtDNA content, which contradicts the findings of May-Panloup et al. (2003) who observed an increase in mtDNA content among infertile patients. However, our findings are consistent with those reported by Kasashima et al. (2008) in Hela cells and Zhang et al. (2020) in mice, demonstrating that the knockdown of phb1 leads to a decrease in mtDNA content. The molecular mechanism of the reduction of mtDNA content caused by phb1 knockdown O. tankahkeei spermiogenesis still needs to be further studied, and the effect of phb1 knockdown on mitochondrial generation of related proteins should be detected at the protein level in the future. Right now, due to the lack of antibodies, we won't be able to do that anytime soon. As previously noted, the semi-quantitative RT-PCR technique is associated with several limitations. The quantitative accuracy achievable through this technique is limited, and it can only provide a rough estimation of the expression level of the target gene. Therefore, we employed qPCR to re-examine the alterations in cyt b mRNA expression. The findings revealed that phb1 knockdown could lead to a decrease in mtDNA content. These outcomes have been revised and presented in Figure 11.
  2. We are sorry for the inaccurate description of the expression result of bcl2 mRNA, we have changed it to “……; increased the mRNA expression of the anti-apoptotic gene bcl2 is more stable.”
  3. In this study, we did not investigate the effects of PHB1 interference on spermatid and sperm morphology and function, as these aspects will be addressed in our forthcoming article.

Comments on the Quality of English Language

I am not a native speaker. My comments need to be taken with caution. However, I do see multiple instances where the English is not sufficiently clear and easy to read. So I recommend a professional editing service. Below I give only few examples (there are more to correct than what is listed below!)

 Answer: Thank you for this valuable suggestion. We have submitted the revised manuscript to Editage for grammatical refinement.

"Further, Ot-PHB1 and Ot-PHB2 were highly colocalized, suggesting they may function primarily as Ot-PHBs in O. tankahkeei. Ot-PHB1 proteins were mainly expressed and localized in mitochondria during spermiogenesis, implying that they may function in mitochondria" (lines 18-21)

These two sentences do not have any meaning. If I can be allowed an analogy, this is like saying I saw the rat 1 and the rat 2 together, so they probably function as rats. I saw them in a sewage tube, so they probably live in the sewage tube.

 Answer: The error has been rectified.

"colocated" (line 21) colocalized

  Answer: The error has been rectified.

(mitochondria) "store energy for sperm motility" (line 54) provide

Answer: The error has been rectified in line 58.

"studying the sperm shape of O. tankahkeei can provide a theoretical basis for artificial breeding" (line 73-4) "shape" should be omitted

 Answer: The error has been rectified.

"look up SPFH domain of PHBs I" (lines 115-6) unintelligible

Answer: We have changed it to “search for the SPFH domain in Ot-PHBs”.

Reviewer 2 Report

The manuscript by Wang and colleagues entitled “The effect of prohibitins on mitochondrial function during Octopus tankahkeei spermiogenesis” overviews a detailed and thorough set of experiments to identify prohibitin 1 and 2 functions in the mitochondria during spermatid and spermatozoa development in an economically important species. The study is overall well-written and conclusions appear sound, with a particular note to the diverse set of techniques the authors use to characterize effects. This reviewer has mostly minor comments that should be addressed prior to further consideration for publication.

Major comment:

The knockdown of phb1 and resulting mitochondrial effects on ROS, mtDNA content, and apopotic genes are interesting. Considering that the authors also note cytoplasmic presence of PHB2 and possible other roles, was any consideration given to those knockdown experiments as well? Do the authors speculate that a greater diversity of effects may be observed, or would other methods be preferred to more closely investigate PHB2-specific roles? This may be worth a further mention in the Discussion.

Minor comments:

Line 17. Please define SPFH here and also in Line 38.

Line 42. Do the authors mean PHBs here? If so, please be consistent between PHBS and PHBs.

Line 52. Please rephrase this to avoid the word “evolve” in this sentence.

Line 76-77. Please adjust this sentence to “In stage II of the spermatid…” and change as needed throughout similar sentences.

Line 89. The green color (mitochondria) and abbreviated spermatid structures should be defined in the figure legend.

Line 94. How were samples preserved prior to RNA extractions? What were the sample sizes per tissue?

Line 112-118. This reviewer understand the point of this sentence, but it is very confusing to read and difficult to parse. Perhaps separate this into two sentences, or rephrase.

Table 1. Please list PCR product sizes for these primer sets.

Lines 127-131. More explanation is needed regarding qPCR methodology. Were PCR efficiencies checked for each gene and validated? Please state that B-actin was used as a reference gene, and was this checked for stability across tissues? In this reviewer’s experience (not in O. tankahkeei), finding an appropriate reference gene across multiple tissues with such divergent expressional profiles can be difficult.

Fig. 2. Please define in the legend for the audience the meanings of the red and green boxes in A (start and stop codons). What do the red and blue arrows in B refer to? Please also rephrase the descriptions of D, E, and F for clarity. As written, it seems like F is indicating a tertiary structure diagram. These comments are also relevant to Fig. 3.

Figs. 2 and 3. Why is Octopus bimaculoides not in the phylogenetic trees if this is discussed in the relevant Results text? Please consider revising this for clarity.

Fig. 4. Was either semi-quantitative PCR or qPCR assessed for statistical significance across tissues? Please consider adding this.

Line 259 and Fig. 5C. The rationale for this pattern of PHB1 and PHB2 could use some additional explanation in the Results text, or perhaps move this to a detailed Discussion section? For this reviewer, some additional explanation and context is needed to better understand this.

Fig. 6. Please define SHDA in the figure legend.

Fig. 7. Please define the abbreviations used in stage V here, as well as in other relevant figures.

Fig. 11. This does not appear to be clearly significant to this reviewer, based solely on looking at the gel and graph. What were the sample sizes here per treatment? Please add this to the methods and consider some additional details regarding sample sizes throughout. Could the authors explain this further in Lines 319-320?

Lines 325-326. Please consider revising the explanation regarding blc2 here. There was no significant increase in expression, so phrasing this as more stable or not different may be more appropriate.

Line 354. The end of this sentence is repetitive with the earlier part.

Author Response

Reviewer 1

The manuscript by Wang and colleagues entitled “The effect of prohibitins on mitochondrial function during Octopus tankahkeei spermiogenesis” overviews a detailed and thorough set of experiments to identify prohibitin 1 and 2 functions in the mitochondria during spermatid and spermatozoa development in an economically important species. The study is overall well-written and conclusions appear sound, with a particular note to the diverse set of techniques the authors use to characterize effects. This reviewer has mostly minor comments that should be addressed prior to further consideration for publication.

Major comment:

The knockdown of phb1 and resulting mitochondrial effects on ROS, mtDNA content, and apopotic genes are interesting. Considering that the authors also note cytoplasmic presence of PHB2 and possible other roles, was any consideration given to those knockdown experiments as well? Do the authors speculate that a greater diversity of effects may be observed, or would other methods be preferred to more closely investigate PHB2-specific roles? This may be worth a further mention in the Discussion.

Answer: Thank you for this valuable suggestion. We further explore the potential function of phb2, which is located in cells excluding mitochondria, in paragraph 3 of the discussion section. Just as you suggested, based on this study, we will investigate the role of phb2 in O. tankahkeei spermiogenesis from two perspectives in the future:

(1) To elucidate the mechanism underlying PHB2 translocation from the nucleus to mitochondria during O. tankahkeei spermiogenesis.

(1.1) We further observed the localized of PHB2 protein during O. tankahkeei spermiogenesis using laser confocal microscopy and confirmed its migration from the nucleus to mitochondria.

(1.2) If migration occurs, the mechanism of PHB2 localization and migration can be investigated by constructing truncated mutants and amino acid mutants of the PHB2 sequence. The expression and localization of each mutant can then be observed using laser confocal microscopy to determine which sequence is responsible for the localization and migration of PHB2 to mitochondria.

(2) To further elucidate the role of PHB2 in maintaining mitochondrial morphology and function stability during O. tankahkeei spermiogenesis.

 (2.1) Similar to phb1, we silenced Ot-phb2 and examined its impact on ROS levels, mtDNA content, and apoptotic genes expression.

 (2.2) (The study revealed that OPA1, a key player in mitochondrial fusion and remodeling, represents the primary target of PHB2 within mitochondria. Deletion of PHB2 results in selective loss of OPA1, leading to severe mitochondrial dysfunction, apoptosis and aberrant cristae morphogenesis (Kowno et al, 2014).

(2.2.1) We will investigate the interaction between PHB2 and OPA1 using immunoprecipitation technology.

(2.2.2) After the knockdown of phb2, we will investigate changes in opa1 mRNA and OPA1 protein expression using qPCR and western blot analysis..

(2.2.3) After the knockdown of phb2, we observed alterations in mitochondrial morphology as revealed by electron microscopy.……….

Sievers L, Billig G, Gottschalk K, et al.  Prohibitins are required for cancer cell proliferation and adhesion[J]. PLoS One, 2010, 5(9): e12735. 

Kowno M, Watanabe-Susaki K, Ishimine H, et al. Prohibitin 2 regulates the proliferation and lineage-specific differentiation of mouse embryonic stem cells in mitochondria[J]. PLoS One, 2014, 9(4): e81552. 

Minor comments:

Line 17. Please define SPFH here and also in Line 38.

Answer: We have redefined SPFH in line 20-21 and line 41.

Line 42. Do the authors mean PHBs here? If so, please be consistent between PHBS and PHBs.

Answer: We have revised PHBS to PHBs and thoroughly reviewed the entire text to prevent recurrence of similar errors.

Line 52. Please rephrase this to avoid the word “evolve” in this sentence.

Answer: We have changed this sentence to “The morphology, gene expression, protein composition, and migration of mitochondria” in line 56-57.

Line 76-77. Please adjust this sentence to “In stage II of the spermatid…” and change as needed throughout similar sentences.

Answer: We have changed “In stage II of the spermatid” to “In stage II of spermiogenesis”, and made similar changes throughout the full text.

Line 89. The green color (mitochondria) and abbreviated spermatid structures should be defined in the figure legend.

Answer: We have revised the caption of Figure 1 to better reflect its significance in our research

Line 94. How were samples preserved prior to RNA extractions? What were the sample sizes per tissue?

Answer: We apologize for the lack of clarity in this section and have included relevant information in Section 2.1 to address this issue..

Line 112-118. This reviewer understand the point of this sentence, but it is very confusing to read and difficult to parse. Perhaps separate this into two sentences, or rephrase.

Answer: We apologize for any confusion caused by the unclear wording in section 2.3 and have since revised it to ensure clarity.

Table 1. Please PCR product sizes for these primer sets.

Answer: We have included the sizes of PCR products amplified by these primer sets in Table 1.

Lines 127-131. More explanation is needed regarding qPCR methodology. Were PCR efficiencies checked for each gene and validated? Please state that B-actin was used as a reference gene, and was this checked for stability across tissues? In this reviewer’s experience (not in O. tankahkeei), finding an appropriate reference gene across multiple tissues with such divergent expressional profiles can be difficult.

Answer: Thank you for this valuable suggestion. (1) We've added more explanation needed for qPCR methodology in section 2.4. (2) We have verified the amplification efficiency of the primers, the result indicated that they can be used for qPCR. We will upload the excel that calculate the amplification efficiency of the primers. (3) We went through a lot of literature and found that the β-actin gene was often used as an internal control in reproductive biology research (Qin et al., 2018; Jia et al., 2018; Ni et al., 2019 etc). In addition, Wang et al. (2010), Mao et al. (2012) and Long et al. (2015) used β-actin as a reference gene for the study of gonad-development-related genes expression in Octopus tankahkeei. Therefore, we choose β-actin as a internal control. This is a great suggestion. In future studies, we will consider this recommendation and explore whether β-actin can be used as an reference gene for the study of gonad-development-related genes expression.

Qin G. Luo W. Tan S. et al. 2018. Dimorphism of sex and gonad-development-related genes in male and female lined seahorse, Hippocampus erectus, based on transcriptome analyses. Genomics, https://doi:10.1016/j.ygeno.2018.11.008. 

Jia Y. Zheng J. Chi M. et al. 2018. Molecular identification of dmrt1 and its promoter CpG methylation in correlation with gene expression during gonad development in Culter alburnus. Fish Physiology and Biochemistry, https://doi:10.1007/s10695-018-0558-1 

Ni F. Yu H. Liu Y. et al. 2019. Roles of piwil1 gene in gonad development and gametogenesis in Japanese flounder, Paralichthys olivaceus. Gene.  https://doi:10.1016/j.gene.2019.03.045. ),

Wang et al (2010), Mao et al (2012) and Long et al (2015) used β-actin as an reference gene for the study of gonad-development-related genes expression in Octopus tankahkeei.

Long LL, Han YL, Sheng Z, Du C, Wang YF, Zhu JQ. Expression analysis of HSP70 in the testis of Octopus tankahkeei under thermal stress. Comp Biochem Physiol A Mol Integr Physiol. 2015 Sep;187:150-9. doi: 10.1016/j.cbpa.2015.05.022. Epub 2015 May 30. PMID: 26033497.

Mao HT, Wang DH, Lan Z, Zhou H, Yang WX. Gene expression profiles of prohibitin in testes of Octopus tankahkeei (ot-phb) revealing its possible role during spermiogenesis. Mol Biol Rep. 2012 May;39(5):5519-28. doi: 10.1007/s11033-011-1355-4. Epub 2011 Dec 21. PMID: 22187346.

Wang W, Zhu JQ, Yang WX. Molecular cloning and characterization of KIFC1-like kinesin gene (ot-kifc1) from Octopus tankahkeei. Comp Biochem Physiol B Biochem Mol Biol. 2010 Jul;156(3):174-82. doi: 10.1016/j.cbpb.2010.03.004. Epub 2010 Mar 19. PMID: 20304088.

Fig. 2. Please define in the legend for the audience the meanings of the red and green boxes in A (start and stop codons). What do the red and blue arrows in B refer to? Please also rephrase the descriptions of D, E, and F for clarity. As written, it seems like F is indicating a tertiary structure diagram. These comments are also relevant to Fig. 3.

Answer: We have revised the caption of Figure 2 and Figure 3 to better reflect its significance in our research.

Figs. 2 and 3. Why is Octopus bimaculoides not in the phylogenetic trees if this is discussed in the relevant Results text? Please consider revising this for clarity.

Answer: We have revised the phylogenetic trees presented in Figure 2 and Figure 3.

Fig. 4. Was either semi-quantitative PCR or qPCR assessed for statistical significance across tissues? Please consider adding this.

Answer: We modified Figure 4 and added statistical significance across tissues.

Line 259 and Fig. 5C. The rationale for this pattern of PHB1 and PHB2 could use some additional explanation in the Results text, or perhaps move this to a detailed Discussion section? For this reviewer, some additional explanation and context is needed to better understand this.

Answer: We have added relevant information in the legend of Figure 5 and in the first paragraph of the discussion section.

Fig. 6. Please define SDHA in the figure legend.

Answer: We have provided a definition of SDHA in the legend of Figure 6.

Fig. 7. Please define the abbreviations used in stage V here, as well as in other relevant figures.

Answer: We have defined the abbreviations used in stage V in Figures 7 and 8.

Fig. 11. This does not appear to be clearly significant to this reviewer, based solely on looking at the gel and graph. What were the sample sizes here per treatment? Please add this to the methods and consider some additional details regarding sample sizes throughout. Could the authors explain this further in Lines 319-320?

Answer: Thank you for this valuable suggestion. We have added relevant information into the methods and results sections.

Lines 325-326. Please consider revising the explanation regarding blc2 here. There was no significant increase in expression, so phrasing this as more stable or not different may be more appropriate.

Answer: Thank you for this valuable suggestion. We have changed it to “….; the mRNA expression of the anti-apoptotic gene bcl2 is more stable (Fig. 12b).”

Line 354. The end of this sentence is repetitive with the earlier part.

Answer: Thank you for this valuable suggestion. We have deleted the sentence.

Round 2

Reviewer 1 Report

This is a second review of the manuscript on prohibitins 1 and 2 in spermiogenesis of Octopus tankahkeei. The presented revised manuscript did improve considerably compared to the original submission; however it requires further revision before it could be ready for publication. The issue of the statistical significance of the findings of the RNAi experiment is critical and must be addressed (see the specific remarks below):

Reevaluating the remarks and responses

section 2.3

This section was improved but the methods used are still not properly cited, as requested. English needs to be watched (e.g. complimentary/complementary).

section 2.4 and 2.8 qPCR

The authors cannot demonstrate that beta actin is relevant endogenous control gene for the tissue and species. However, it was at least used in past in testis of the Octopus tankahkeei, so at least there is the possibility of comparison. Past publications using beta actin as the endogenous control in Octopus tankahkeei should be cited.

sections 2.5 and 2.6 (antibodies)

The statement from the authors response "The band exhibited singularity and corresponded to the target protein's size." should in some form occur in methods. Specificity of the used antibodies is paramount in experiments of this type.

section 3.4

the study as a whole would be much more sound if the authors did already complete the experiments that they suggest in the response to my remark.

Section 3.7-10 and figure 9-12.

I am still unable to assess the statistical significance of the results as there is no indication what is the meaning of the error bars. I have to insist that I require here revision of the statistical tests to ascertain that the significance is duly assigned.

As an example, I can use figure 10 (hydrogen peroxide) where the error bars clearly overlap, therefore there can be no statistical difference whatever meaning the error bars have. However, the problem is not limited to this particular figure and the extent of the problem cannot be assessed accurately without knowing the meaning of the error bars.

English in my opinion (of a non-native speaker) has improved very much compared to the original submission; however, some new errors have been introduced during the editing (one example is above).

Author Response

This is a second review of the manuscript on prohibitins 1 and 2 in spermiogenesis of Octopus tankahkeei. The presented revised manuscript did improve considerably compared to the original submission; however it requires further revision before it could be ready for publication. The issue of the statistical significance of the findings of the RNAi experiment is critical and must be addressed (see the specific remarks below):

Reevaluating the remarks and responses

section 2.3

This section was improved but the methods used are still not properly cited, as requested. English needs to be watched (e.g. complimentary/complementary).

Answer: Thank you for your valuable suggestion. Table 2 is included to enumerate the online tools utilized for protein structure analysis, with corresponding literature citations.

section 2.4 and 2.8 qPCR

The authors cannot demonstrate that beta actin is relevant endogenous control gene for the tissue and species. However, it was at least used in past in testis of the Octopus tankahkeei, so at least there is the possibility of comparison. Past publications using beta actin as the endogenous control in Octopus tankahkeei should be cited.

Answer: We have made modifications in accordance with your suggestion.

sections 2.5 and 2.6 (antibodies)

The statement from the authors response "The band exhibited singularity and corresponded to the target protein's size." should in some form occur in methods. Specificity of the used antibodies is paramount in experiments of this type.

Answer: We have included relevant information in section 2.6.

section 3.4

the study as a whole would be much more sound if the authors did already complete the experiments that they suggest in the response to my remark.

Answer: Thank you for your affirmation of the content we responded to. For a long time to come, we will work on these experiments one by one and try to complete them. After all, they are challenging for us. We will put these findings and the effects of phb1and phb2 interference on spermatid and sperm morphology and function in the next article to explore how do phb1 and phb2 regulate Octopus tankahkeei spermiogenesis. Moreover, the findings of present study can offer supporting evidence for the hypothesis that PHBs play a role in Octopus tankahkeei spermiogenesis and serve as a foundation for further research.

Section 3.7-10 and figure 9-12.

I am still unable to assess the statistical significance of the results as there is no indication what is the meaning of the error bars. I have to insist that I require here revision of the statistical tests to ascertain that the significance is duly assigned.

As an example, I can use figure 10 (hydrogen peroxide) where the error bars clearly overlap, therefore there can be no statistical difference whatever meaning the error bars have. However, the problem is not limited to this particular figure and the extent of the problem cannot be assessed accurately without knowing the meaning of the error bars.

Answer: In the experiment, a single measurement will inevitably produce errors. Therefore, we conducted three measurements and used the average value to represent the measured amount. We also used error bars to characterize the distribution of data, where the height of each error bar represents ± standard error. This is equivalent to calculating standard deviation. A large standard deviation indicates a significant difference between the majority of values and their mean, while a smaller standard deviation suggests that these values are closer to the average. The formula for the standard deviation is

The error bars in Figure 10 do not overlap, however, it should be noted that - standard error bars were intentionally omitted during the drawing process.

We present the results of our significance calculations using SPSS software in the PDF document titled “Significance”.

Comments on the Quality of English Language

English in my opinion (of a non-native speaker) has improved very much compared to the original submission; however, some new errors have been introduced during the editing (one example is above).

Answer: We carefully reviewed the entire text to prevent any potential spelling and grammar errors from recurring.